# Unraveling the Impact of Intratumoral Heterogeneity on EGFR Tyrosine Kinase Inhibitor Resistance in *EGFR*-Mutated NSCLC

**DOI:** 10.3390/ijms24044126

**Published:** 2023-02-18

**Authors:** Keigo Kobayashi, Aaron C. Tan

**Affiliations:** 1Division of Medical Oncology, National Cancer Centre Singapore, Singapore 169610, Singapore; 2Duke-NUS Medical School, National University of Singapore, Singapore 169857, Singapore

**Keywords:** EGFR-TKI, intratumoral heterogeneity, drug-tolerant persister cells, extrachromosomal DNA, concurrent genetic alterations

## Abstract

The advent of tyrosine kinase inhibitors (TKIs) for treating epidermal growth factor receptor (EGFR)-mutated non-small-cell lung cancer (NSCLC) has been a game changer in lung cancer therapy. However, patients often develop resistance to the drugs within a few years. Despite numerous studies that have explored resistance mechanisms, particularly in regards to collateral signal pathway activation, the underlying biology of resistance remains largely unknown. This review focuses on the resistance mechanisms of EGFR-mutated NSCLC from the standpoint of intratumoral heterogeneity, as the biological mechanisms behind resistance are diverse and largely unclear. There exist various subclonal tumor populations in an individual tumor. For lung cancer patients, drug-tolerant persister (DTP) cell populations may have a pivotal role in accelerating the evolution of tumor resistance to treatment through neutral selection. Cancer cells undergo various changes to adapt to the new tumor microenvironment caused by drug exposure. DTP cells may play a crucial role in this adaptation and may be fundamental in mechanisms of resistance. Intratumoral heterogeneity may also be precipitated by DNA gains and losses through chromosomal instability, and the role of extrachromosomal DNA (ecDNA) may play an important role. Significantly, ecDNA can increase oncogene copy number alterations and enhance intratumoral heterogeneity more effectively than chromosomal instability. Additionally, advances in comprehensive genomic profiling have given us insights into various mutations and concurrent genetic alterations other than EGFR mutations, inducing primary resistance in the context of tumor heterogeneity. Understanding the mechanisms of resistance is clinically crucial since these molecular interlayers in cancer-resistance mechanisms may help to devise novel and individualized anticancer therapeutic approaches.

## 1. Introduction

Lung cancer is the leading cause of cancer-related deaths globally [1]. The identification of epidermal growth factor receptor (*EGFR*) mutations as a driver oncogene in non-small-cell lung cancer (NSCLC) and the efficacy of EGFR-tyrosine kinase inhibitors (TKIs) in *EGFR*-mutated patients has revolutionized lung cancer treatment, with responses observed in 60–80% of patients [2,3,4,5,6,7,8,9,10].

Approximately 90% of the most commonly occurring mutations in the EGFR gene are the classical activating mutations of exon 19 deletions and exon 21 L858R point mutations, which lead to the continuous activation of EGFR [11]. The most frequently observed uncommon EGFR mutations include G719X, S768I, L861Q, exon 20 insertions, and compound mutations. Uncommon EGFR mutations are any other mutations in the EGFR gene that are not as commonly seen as the classical mutations. These mutations may have differing effects on the function of the EGFR protein and may also respond differently to EGFR-targeted drug treatments [12].

EGFR-TKIs have so far been developed and utilized in the clinical setting as below: first generation (gefitinib and erlotinib), second generation (afatinib and dacomitinib), and third generation (osimertinib). The “second-generation” EGFR-TKIs earned their name due to their irreversible pan-HER inhibitory activity. “Third-generation” inhibitors target the T790M mutation, which is the most frequent resistance mechanism to both first- and second-generation EGFR-TKIs, specifically [13].

However, almost all EGFR TKI responders acquire drug resistance within a few years, with median progression-free survival ranging from 9.2 to 18.9 months for first-, second-, and third-generation EGFR-TKIs. Many studies have revealed several acquired-resistance mechanisms and developed therapeutic strategies countervailing them [4,5,6,7,8,9,10]. Third-generation EGFR-TKIs can, for instance, further induce on-target resistance mutations, such as the *EGFR* C797S mutation [14]. Bypass signaling pathway activation, including *MET* and *HER2* amplification or histologic transformation (small cell or squamous cell transformation), are significant resistance mechanisms to all generations of EGFR-TKIs [15,16,17]. As a result, there is ongoing research and development of fourth-generation EGFR-TKIs [18,19].

In clinical practice, although tumor shrinkage is observed with EGFR-TKIs, complete radiologic tumor disappearance is incredibly rare, and in most cases, residual radiologic disease remains. It is largely unknown why tumors persist despite the potent tumor-suppressive effects of molecular-targeted agents.

One hypothesis is intratumoral cellular heterogeneity. There may be various subclones of cancer cells, including those that are either sensitive or resistant to EGFR-TKIs within a single tumor. Thus, not every cancer cell can be eliminated through a single therapy targeting specific molecules. It is well established that treatment-naïve tumors already exhibit significant heterogeneity [20].

On the other hand, even in studies using lung cancer cell lines that are considered relatively homogeneous, EGFR-TKIs fail to eradicate all cancer cells [21]. This suggests that there are other mechanisms besides tumor heterogeneity that prevent tumor eradication. Previous studies have revealed that suppressing the main driver oncogene signaling pathway can lead to the complementary activation of other collateral bypass signaling pathways [22].

This review focuses on understanding resistance to EGFR-TKI therapy from both basic and translational research perspectives, with a specific emphasis on intratumoral heterogeneity. It provides an overview of the current understanding and future prospects in this field.

## 2. Collateral Signal Pathway Activation

Three oncogenic pathways that promote cell growth and division are: PI3K/AKT/mTOR, RAS/RAF/ERK MAPK, and STAT/JAK. EGFR signals through the PI3K and RAS/RAF/ERK MAPK pathways. Bypass tracks may eventually result in aberrant activation of the downstream pathway or associated signaling pathways under targeted therapy. This will guarantee that aberrant signaling persists despite ongoing target inhibition. Notably, bypass tracks are expected to form more frequently when more potent target inhibitors are utilized [23].

The simultaneous activation of multiple receptor tyrosine kinases (RTKs) is a frequent occurrence among oncogenic drivers following RTK inhibition.

Notably, EGFR inhibition in NSCLC may result in the emergence of MET, HGF, and ERBB2 amplification, AXL activation, FGF2–FGFR1 loop mutations, IGF1R activation, and fusion events.

MET amplification, which occurs in 4–10% of cases of resistance, stimulates downstream MEK/ERK and PI3K/Akt signaling pathways by phosphorylating ERBB3 [24,25,26,27]. EGFR-mutant NSCLC patients may have MET amplification prior to EGFR-TKI treatment, and EGFR-TKI exposure may increase the rate of MET amplification and activate the bypass pathway through transactivated ERBB3 [28].

HGF, which triggers the MET-HGF complex, also stimulates downstream MEK/ERK and PI3K/Akt signaling pathways through Gab1 as an adapter protein [29]. HGF overexpression has been shown to be commonly detected in NSCLC patients both before and after EGFR-TKI treatment (23/97, 24%, and 45/97, 46%, respectively) [30].

ERBB2 overexpression is a mechanism of acquired resistance for NSCLC patients without the EGFR T790M mutation after EGFR-TKI treatment, potentially initiated by ERBB3 which strongly activates downstream PI3K/Akt signaling [31,32,33,34,35]. This is because ERBB3 can bind to the p85 subunit of PI3K and activate PI3K directly [36].

AXL, activated by its ligand Gas6, can stimulate PI3K/Akt, MAPK, or NF-κB signaling and may also be associated with EMT [37,38].

FGFR1 can form an autocrine loop with its ligand FGF2, activating downstream PI3/Akt signaling [39].

IGF1R activation by IGF-IGF1R binding can also contribute to EGFR-TKI resistance, but its activation can be regulated by IGFBP-3 and IGFBP-4, which are active regulators of the six high-affinity IGFBPs (IGFBP-1 to IGFBP-6) [40,41,42].

Recently, acquired gene fusions after EGFR-TKIs failure in EGFR-mutated NSCLC have been detected among bypass tracks [43]. RET fusions are most frequent (46%), followed by ALK (26%), NTRK1 (16%), and FGFR3 (11%) [43,44,45]. Through the use of whole exome sequencing or RNA sequencing on tissue samples, these gene fusions were found to occur at higher rates after the use of third-generation EGFR-TKIs, compared with first- or second-generation EGFR-TKIs [44].

Another potential mechanism of collateral signal pathway activation is via integrins. Integrins are transmembrane receptors that facilitate cell–cell and cell–extracellular matrix adhesion and may play a vital role in countering EGFR-TKI treatments. Integrins and EGFR share similar signaling pathways, allowing cancer cells to adapt and withstand therapy more effectively [46]. Additionally, exposure to EGFR-TKIs can trigger the activation of integrin β3, boosting AXL expression through the YAP pathway and resulting in greater resistance [47].

## 3. Intratumoral Heterogeneity

Cancer is a highly diverse disease in nature. Recent research in cancer genomics has uncovered that even within a single tumor in a patient, there often exist multiple subclones [48,49]. Multiple subclones presumably accelerate the evolution of tumor resistance to treatment, serving as a potential source of resistant clones [50]. Cancer stem cell clones and their interaction in three-dimensional space may contribute to intratumoral heterogeneity of the genome [51,52].

Furthermore, intratumoral heterogeneity may exist not only in the genome but also in other omics layers, such as the epigenome and transcriptome. The intratumoral heterogeneity in each layer may further correlate with each other [53].

In lung cancer, recent research has indicated that drug-tolerant persister (DTP) cells may exhibit similar characteristics to stem cells, acting at several levels of the omics hierarchy and being involved in various resistance mechanisms (Figure 1).

Intratumoral heterogeneity may also be precipitated by DNA gains and losses through chromosomal instability and extrachromosomal DNA (ecDNA) [54,55]. Specifically, amplification of ecDNA has been found to be more effective in increasing the copy number of oncogenes and promoting intratumoral diversity compared with chromosomal instability [56].

### 3.1. DTP Cells and Resistance Mechanisms

Intratumoral heterogeneity means that even if the majority of treatment-sensitive clones shrink with treatment, a small number of treatment-resistant clones may remain, which may eventually proliferate and lead to recurrence or progression in lung cancer [57,58]. According to recent findings, a tiny cluster of lung cancer cells has the potential to evade the effects of anti-cancer drugs by entering into a temporary phase of slow growth referred to as the DTP state [59,60,61]. This reversible condition can empower cancer cells to endure medication treatment before eventually acquiring alterations and proliferating, which phenotypically manifests as acquired resistance. Without therapeutic selective pressure, these DTP cells could subsequently be reconverted as naïve cells if the drug is discontinued (Figure 1).

Several genetically acquired alterations, such as *EGFR* T790M or *MET* amplification, may be obtained during the DTP state. A conformational shift in the kinase known as *EGFR* T790M lowers drug affinity for the target. Target amplification, such as MET amplification, may also reduce the drug’s potency by outpacing its capacity for inhibition. Thus, the DTP state could serve as a repository for the cultivation of drug-resistant cells [62].

Several preclinical studies have revealed that DTP cells adjust to changing environments through several layers of omics: metabolism, transcriptome, epigenome, immunity, and tumor microenvironment (TME) (Table 1).

The metabolism of cancer cells will undergo changes to accommodate the new surroundings brought about by drug exposure. Recent studies reveal that cancer cells consume oxygen through mitochondrial oxidative phosphorylation to an equal or greater extent than normal cells [63]. Especially, DTP cells have an elevated reliance on mitochondrial oxidative phosphorylation, and therefore, they should possess an antioxidant stress defense mechanism to counteract the excessive production of oxygen-containing substances, such as superoxide and lipid peroxide. Thus, DTP cells utilize glutathione peroxidase 4 (GPX4) to detoxicate lipid peroxides. It is certificated in NSCLC cells that GPX4 inhibition can accumulate lipid peroxides and lead to ferroptosis, an oxidative form of cell death [64,65].

Similarly, DTP cells have been associated with high aldehyde dehydrogenase (ALDH) activity, which serves to guard them against the harmful effects of lipid peroxidation [61]. For example, the use of a combination therapy consisting of an ALDH inhibitor (disulfiram) and cisplatin-based chemotherapy resulted in an improvement of survival for advanced NSCLC patients, indirectly suggesting that ALDH inhibition may counter drug resistance [66].

Autophagy, unlike apoptosis, is a mechanism for cell survival by degrading and recycling proteins, other macromolecules, and organelles. Autophagy is vital in eliminating defective mitochondria produced by reactive oxygen species through mitochondrial respiration. The accumulation of damaged mitochondria by drug exposure promotes malignancy by increasing oxidative stress. AXL, a receptor tyrosine kinase, has been linked to autophagy. A recent report indicates that after undergoing erlotinib therapy, AXL-positive DTP cells exhibit resistance to apoptosis due to heightened autophagy. The use of the AXL inhibitor bemcentinib may delay the development of resistance to EGFR-TKI treatments [67].

Epigenetics involves a gene regulatory mechanism unrelated to DNA sequences, such as DNA methylation, hydroxymethylation, and histone protein modifications. The transcriptome is also closely related to epigenetics. For example, dynamic remodeling of the chromatin structure allows epigenetic regulation of apoptosis and gene expression.

A phenotypic feature of DTP cells is their reversible biological capacity, which is not specified by the genome but by reversible epigenomic regulation [59].

EGFR-TKI exposure can increase the expression of the epigenetic regulator H3K4 demethylase, KDM5A. Since KDM5A suppresses the transcriptive activity of H3K4, upregulation of KDM5A expression can lead to drug resistance. KDM5 inhibitors, such as CPI-455, can increase the levels of H3K4 trimethylation and exert anticancer effects in NSCLC [68]. IGF-1R was also reported to interact with KDM5A expression positively. Osimertinib, a third-generation EGFR-TKI, stimulates FOXA1, a transcriptomic factor of IGF-1R, causing an increase in IGF-1R expression. Therefore, combining EGFR TKI and IGF-1R inhibitors may eliminate IGF-1R-dependent KDM5A expression [59,69].

Long interspersed repeat element 1 (LINE-1) retrotransposons are autonomous transposable genetic factors occupying about 17% of the human genome, which can move their DNA sequences. While the transposable factor drives genetic diversity and promotes biological evolution in the long term, it can create DNA mutations leading to malignant disease in a short time. The trimethylation of H3K9, which is most significantly increased over LINE-1, maintains the survival of DTP cells through the mediation of heterochromatin formation. Inhibiting methyltransferases of LINE-1 components such as SETDB1 and G9a leads to DTP cell disruption [70].

DTP cells are protected to survive not only by H3K9-methylation but also by H3K27-methylation [71]. Polycomb repressive complex 2 (PRC2), involved in DNA damage repair, can methylate H3K27 via EZH2, the enzymatically active core subunit of PRC2. Thus, blocking EZH2 activity may reduce the population of DTP cells in NSCLC [72].

Hyperfunction of the tyrosine phosphatase SHP2 has the potential to contribute to the growth of a cancerous tumor. SHP2, with dephosphorylated- YAP and -TAZ, is translocated into the nucleus where it triggers Wnt-β-catenin and YAP- and TAZ-dependent transcriptional activation. Inside the nucleus, the main transcription factor that facilitates the actions of YAP is TEAD, which controls the expression of genes that promote cell growth and prevent cell death [73]. The combination treatment of EGFR and MEK inhibitors induces DTP cells through activation of YAP/TEAD, forming a complex with the transcription factor SLUG that is related to epithelial–mesenchymal transition (EMT), thus suppressing apoptotic pathways [74]. The YAP-TEAD complex also affects AXL, acting as transcriptomic factors that regulate significant genes and determine drug sensitivity [75,76].

The SHP2 signaling pathway activates the Wnt/β-catenin and YAP/TAZ in the nucleus, while EGFR TKIs can trigger the formation of DTP cells through the activation of β-catenin signaling, which is dependent on a non-canonical Notch3 pathway [77]. Genes related to the β-catenin pathway are found to be abundant in DTP cells from lung cancer patients who have received EGFR TKI treatment [78]. It has also been discovered that overexpression of AURKA, a key player in cellular mitosis, and FGFR3, which drives EMT programming, can lead to chromosomal instability and the formation of DTP cells [79,80]. Inhibiting the pathways that control the transcription of Wnt/β-catenin, AURKA, and FGFR3 has the potential to eliminate drug resistance when used in combination with EGFR TKIs.

**Table 1 ijms-24-04126-t001:** DTP cell-specific resistance mechanisms and treatment.

Multi-Omics Layer	Mechanisms	Target	Treatment	Key Drug	Year of FDA Approval	Ref.
Metabolism	Autophagic flux increase	AXL	AXL inhibitor	bemcentinib	Not yet (preclinical)	[67]
	Detoxication of lipid peroxidation	GPX4	GPX4 inhibitor	RLS3	Not yet (preclinical)	[64,65]
		ALDH	ALDH inhibitor + cisplatin	disulfiram	1951	[66]
Transciptome	KDM5A upregulation	IGF-1R	IGF-1R inhibitor+EGFR-TKI	linsitinib	Not yet (preclinical)	[69]
Epigenome	H3K4 demethylation	KDM5A	KDM5 inhibitor	CPI-455	Not yet (preclinical)	[68]
	H3K9 trimethylation	LINE-1	G9a inhibitor	UNC-0638	Not yet (preclinical)	[70]
	H3K27 methylation	EZH2	EZH2 inhibitor	GSK126	Not yet (preclinical)	[71,72]
	Apoptosis downregulation	YAP/TEAD-SLUG	EGFR-TKI+MEK-inhibitor	trametinib	Not yet (preclinical)	[74]
	SHP2, Wnt/β-catenin pathway activation	Wnt/β-catenin	Tankyrase inhibitor+EGFR-TKI	XAV939	Not yet (preclinical)	[77,78]
	Cell cycling upregulation	AURKA	AURKA inhibitor+EGFR-TKI	TC-A2317	Not yet (preclinical)	[79]
	Chromosomal instability	FGFR3	FGFR3 inhibitor+EGFR-TKI	AZD4547	Not yet (preclinical)	[80]

### 3.2. Tumor Microenvironment (TME)

Lung cancer cells interact with various cells in the tumor microenvironment (TME), including cancer-associated fibroblasts (CAFs) and tumor-associated macrophages (TAM), as well as normal cells, leading to intratumoral heterogeneity.

CAFs, a major component of the TME, contribute to drug resistance through secretion of growth factors and chemokines, such as HGF [81]. HGF can activate MAPK and PI3K/AKT pathways through interaction with the MET receptor [25,26]. A crosstalk between cancer cells and CAFs can also activate the IGF-IGF1R pathway along with EMT- and stemness-related signaling [82,83].

TAMs and myeloid-derived suppressor cells (MDSCs) also fill a pivotal role in resistance to EGFR-TKIs, with S100A9-positive MDSCs linked to poor response to EGFR-TKIs in NSCLC patients [84]. Mechanistically, S100A9 upregulates ALDH1A1 expression and activates the retinoic acid (RA) signaling pathway, making the S100A9-ALDH1A1-RA axis a potential therapeutic target. Treatment with a pan-RAR antagonist can effectively inhibit cancer proliferation [85].


*Immunity*


Drug exposure affects tumor and tumor-infiltrating immune cells. EGFR TKIs can lower PD-L1 expression in EGFR-mutant NSCLC cells, making PD-1/PD-L1 inhibitors ineffective for most patients with this type of cancer [86]. Furthermore, EGFR-mutant NSCLC cells with high PD-L1 expression are resistant to gefitinib due to PD-L1 overexpression that can trigger EMT through activation of the TGF-β/Smad pathway [87].

AXL has been linked to resistance to PD-1 inhibition and suppression of proper antigen presentation by the MHC-I [88]. Thus, inhibiting AXL could reorganize the immune-suppressive TME in a more favorable way [89,90].

### 3.3. Extrachromosomal Circular DNA (ecDNA)

EcDNA is also a significant contributor to tumor genetic heterogeneity. It is found in approximately half of all human cancers, including lung cancer [91]. The presence of ecDNA varies depending on the type of cancer and is not commonly seen in normal tissue [56,92,93]. The behavior and structure of ecDNAs are different between various cancer types [94,95]. EcDNAs lead to tumor heterogeneity and drug resistance by promoting the amplification of oncogenes [96,97,98,99,100].

There are 23 pairs of chromosomes in human cells. Some genes, however, can be amplified in extrachromosomal DNA under specific circumstances. EcDNA refers to a particular kind of circular DNA molecule and exists independently of the chromosomal genome. These ecDNAs are closed circular DNA formations, which can be single- or double-stranded, and exist independently of the chromosomes [101].

Since ecDNAs lack centromeres, they gradually disappear during mitosis [102,103]. They are not subject to Mendel’s laws of inheritance and are instead distributed haphazardly to daughter cells. This can lead to one daughter cell receiving multiple copies of the ecDNA oncogene during division, giving it a proliferation advantage [56,104]. EcDNA amplification enhances genetic diversity and promotes heterogeneity. Furthermore, such a distinct genetic process enables cells to evolve more rapidly [105,106].

There are several different and intricate ways that ecDNA contributes to drug resistance to targeted cancer therapies [107,108,109]. Genomic rearrangements resulting in chromosomal abnormalities, such as chromothripsis, are often the cause of cancer cell development. These abnormalities can lead to changes in gene copy numbers, including the loss of important tumor suppressor genes and an increase in genes that contribute to the malignant progression of cells. Amplified extrachromosomal mutations give tumor cells the ability to adapt to changing environments, including those caused by anti-cancer treatments, potentially resulting in resistance to molecularly targeted therapies [108].

An analysis of the genetic makeup of 20 patients with NSCLC revealed the presence of chromothripsis regions, where an abnormal accumulation of single nucleotide variations (SNVs) and structural variations (SVs) were observed on a particular haplotype [110]. Additionally, it is believed that a number of lung cancer driver fusion oncogenes result from chromothripsis [111].

Furthermore, recent investigations suggest that ecDNA hubs, nucleus-based clusters of roughly 10–100 ecDNAs, facilitate intermolecular interactions between enhancers and genes to support oncogene overexpression [112]. Hubs are formed by ecDNAs that encode numerous different oncogenes in various primary tumors and cancer cell types. When spatially clustered with additional ecDNAs, each one has a higher likelihood of transcription of the oncogene. Such cluster formation could result in more robust proliferation and the eradication of additional cancer cells. This suggests that resistance could arise from the amplification of ecDNAs.

### 3.4. Primary Resistance in Intratumoral Heterogeneity

Primary resistance, or early tumor progression, is a common phenomenon where 20–30% of patients are insensitive to EGFR-TKIs from the beginning, despite the more prevalent acquired resistance to tumor-targeted therapies (Table 2) [4,6,7,8,9,10,113,114].

In accordance with the definition of primary resistance, we calculated it by subtraction of ORR from 100 (%).

Primary resistance is largely attributed to the pre-existence of extra genetic changes within the context of tumor diversity. Driver oncogenes such as EGFR in lung cancer somatic mutations possess intense oncogenic potential known as “oncogenic addiction,” leading to significant therapeutic effects from a single molecularly targeted drug. However, some EGFR-TKI therapy-naive NSCLC patients have additional driver alterations coexisting with EGFR mutations, which partly contribute to the intrinsic primary resistance [115]. In these cases, cancer proliferation depends on multiple oncogenes besides the oncogenic EGFR mutation, potentially resulting in a weaker oncogenic potential of EGFR and making EGFR-TKI monotherapy less effective as a front-line treatment. This section will concentrate on the most prevalent concurrent genetic alterations, such as TP53, PIK3CA, and PTEN, and concurrent driver gene changes, including ALK rearrangement in EGFR-mutant lung cancer patients.


*TP53*


p53 can trigger cell cycle arrest, senescence, and apoptosis [116]. Approximately 35–55% of NSCLC patients have mutations in the TP53 gene, which encodes p53 and has a strong connection to smoking [117]. TP53 is also the most frequent co-alteration seen in EGFR-mutant NSCLC patients, present in 55–65% of such cases [20,118,119].

Clinical outcomes of EGFR-TKI treatments are negatively affected by TP53 status [120]. In NSCLC cell lines with wild-type p53, the drug gefitinib can activate apoptosis by increasing the presence of Fas at the cell membrane and restoring caspase activation, leading to increased TKI sensitivity. On the other hand, it has been observed that cells with mutated p53 have reduced gefitinib-induced apoptosis, resulting in primary resistance [121]. It is worth mentioning that the prognostic impact may also vary based on the specific type of TP53 mutation [122].


*PIK3CA Mutation*


The gene PIK3CA encodes the catalytic subunit of PI3K and its mutations activate the PI3K/AKT pathway [123]. These mutations are uncommon oncogenic drivers, occurring in 2–5% of NSCLC patients, with the E545K and H1047R mutations being the most frequent [124,125,126].

PIK3CA mutations are frequently associated with other oncogenic mutations, particularly EGFR and KRAS [124,127,128]. In fact, they have been found in roughly 3.5% of patients with EGFR mutations, and are linked to intrinsic primary resistance to EGFR-TKIs [129,130]. In a preclinical study, the presence of an activated PIK3CA mutation (p.E545K) in HCC827 cells that already had an EGFR 19 deletion led to resistance to gefitinib [131]. Additionally, a recent study showed that various PIK3CA mutations can impact the prognosis of EGFR-mutant NSCLC patients treated with EGFR-TKIs. Mutations in the p85 binding domain (R88Q, R108H, and K111E) were linked to improved PFS, while mutations in the kinase domain (Y1021H and H1047R), helical domain (E542K), and C2 domain (N345K) correlated with a worse PFS [132]. As a result, the presence of certain PIK3CA mutations can affect the primary resistance to EGFR-TKI therapy.


*PTEN alterations*


Phosphatase and tensin homolog deleted on chromosome 10 (PTEN) is a tumor suppressor that regulates various cellular processes, including cell proliferation, survival, growth, metabolism, migration, and apoptosis [133,134,135]. Its inactivation, which is crucial in lung cancer oncogenesis and progression, leads to the upregulation of the PI3K/AKT pathway and increased tumor progression, resulting in poor EGFR-TKI sensitivity in NSCLC patients [136,137,138,139]. Loss of PTEN is present in over 40% of NSCLC patients, while PTEN mutations are uncommon, occurring in only 2–5% of NSCLC cases [140,141].

PTEN is crucial in controlling the internalization and degradation of EGFR through its regulation of endocytic trafficking [142]. After the EGF ligand binds to the EGFR receptor, the complex is internalized and transported through clathrin-dependent vesicles to early endosomes for sorting [143,144]. PTEN plays a crucial role in the movement of the EGF/EGFR complex from early to late endosomes through phosphoregulation of Rab7 and endosome maturation. This transition allows for the efficient degradation of the receptor, making PTEN inactivation a hindrance to EGFR signaling [144,145,146].


*Concurrent Driver Gene Alterations—ALK Rearrangement*


Anaplastic lymphoma kinase (ALK) rearrangements are a known oncogenic driver found in about 5% of NSCLC patients [147]. The fusion of EML4 with ALK has been shown to have potent oncogenic effects.

Some studies have reported that between 3.9% and 13.6% of EGFR-mutant NSCLC patients also have concurrent ALK rearrangements [148,149]. This simultaneous presence of mutations may result in primary resistance to EGFR-TKIs, but the efficacy of these drugs in such double-positive patients remains controversial with conflicting results [149,150,151].

The coexistence of EGFR mutations and ALK rearrangements in NSCLC remains a subject of debate, with theories suggesting that the two alterations may either exist in different cells within the same tumor or in the same cells [152,153]. The outcome of EGFR-TKI treatment can be influenced by the dominant driver clone in these cases. Determining the efficacy of EGFR-TKIs by examining phosphorylation levels and downstream proteins remains a controversial topic in the clinical field [149,154,155].

## 4. Discussion

Improved understanding of the molecular factors involved in cancer resistance could result in the creation of new, personalized, molecularly-targeted anti-cancer drugs. This review primarily focuses on resistance mechanisms from the viewpoint of intratumoral heterogeneity.

Interestingly, several oncogenic drivers activate the same pathways. For example, IGF1R, FGFR2, and ERBB2 signal through the PI3K and MAPK pathways, such as EGFR. Despite the seemingly diverse driver oncogenes, bypass pathways tend to be similar due to the common targeting of signaling pathways. As a result, a wealth of knowledge has been gained on collateral signaling pathways.

However, the mechanism of intratumoral heterogeneity is varied and mostly unknown. Intratumoral heterogeneity is partly produced by the branching clonal evolution of cancer cells. Multiple subclones presumably accelerate the evolution of tumor resistance to treatment, serving as a source of resistant clones [50].

Recently, several key discoveries have been made about intratumoral heterogeneity. For example, DTP cells can promote the evolution of cancer and neutral evolution, resulting in resistance to molecularly targeted therapies in NSCLC. This is attributed to two common DTP characteristics: slow cell cycling and proliferative activity, and reversible biology, where DTPs can regain proliferation and drug sensitivity after drug withdrawal [59,156,157].

It is crucial to note that the abundance of neutral mutations in a tumor can contribute to its robustness and ability to evolve. After prolonged exposure to therapy that surpasses a certain threshold of DTPs and alters the local microenvironment, some of the many neutral mutations may become driver genes, providing resistance to treatment [158]. The high mutation rate could be a factor that boosts the number of neutral mutations. Genetic instability causes an increase in the mutation rate prior to branching evolution, as seen by changes in mutational signatures over time [53].

Chromosomal instability may also lead to DNA gains and losses in cancer, which contribute to the development of the disease, its progression, and resistance to treatment. This results in intratumoral heterogeneity [54,55,159,160,161]. The cancer genome undergoes fluctuations in DNA structure and quantity, resulting from chromosomal instability. This instability may lead to ongoing changes in somatic copy numbers, with evidence suggesting that these changes frequently happen simultaneously during tumor evolution [162,163,164]. This variability in somatic copy number alterations (SCNAs) can contribute to tumor growth and intratumoral heterogeneity [162,165,166]. A study examining multi-sample phasing and SCNA analysis in 22 different types of tumors, including NSCLC, demonstrates that persistent chromosomal instability drives extensive SCNA heterogeneity [164].

A study has shown that amplifying ecDNA leads to a greater increase in oncogene copy number and intertumoral diversity compared with amplifying chromosomes [56]. In glioblastoma models, the absence of EGFRvIII on ecDNA caused resistance to EGFR TKI drugs, but when the drug was discontinued, EGFR reappeared on ecDNA. This highlights ecDNA’s role in regulating EGFRvIII levels according to the tumor’s demands [164,167].

The amplification of either mutant or wild-type alleles may impair the effectiveness of treatment by surpassing its inhibition capacity, as seen with EGFR amplification after EGFR-TKIs in NSCLC [27,168]. The relationship between ecDNA and increased EGFR amplification after EGFR-TKI treatment in NSCLC needs further exploration and evidence. The limited understanding of ecDNA stems from its large size, making it difficult to analyze. A recent study used CRISPR-CATCH and in vitro CRISPR-Cas9 treatment, along with pulsed-field gel electrophoresis of agarose-entrapped genomic DNA, to isolate megabase-sized human ecDNAs [169]. Urgent need exists for new research tools to investigate the amplification of oncogenes through ecDNA and understand ecDNA’s role in cancer development, as it may significantly contribute to cancer’s accelerated evolution [170].

The advent of comprehensive genomic profiling has improved our understanding of various mutations and concurrent genetic alterations beyond EGFR mutations [118,171,172]. While acquired resistance to cancer therapies is common, it is crucial to also focus on primary resistance linked to concurrent genetic alterations in light of tumor heterogeneity.

Recently, the development of fourth-generation EGFR-TKIs has gained traction [18,19]. These drugs target resistance mechanisms, such as the EGFR C797X mutations commonly seen in patients who have developed resistance to third-generation EGFR TKIs. Despite their promising potential, these TKIs still face several challenges in effectively targeting on-target resistance mechanisms. One of the biggest obstacles is the potential for the emergence of drug resistance, which could result from off-target mutations or alternative signaling pathway activation. Furthermore, the optimal method of using fourth-generation EGFR TKIs in conjunction with other treatments, such as immunotherapy or chemotherapy, is yet to be determined and requires further research to fully understand their ability to manage intratumoral heterogeneity.

A thorough comprehension of tumor evolution has the potential to enhance lung cancer prognosis. Addressing treatment resistance is a crucial goal, and incorporating knowledge of DTP cells and ecDNA and their impact on intratumoral heterogeneity may lead to more effective therapeutic strategies. The use of comprehensive genomic profiling should be standard, and a deeper understanding of resistance mechanisms can be accelerated through the advancement of precision medicine.

## Figures and Tables

**Figure 1 ijms-24-04126-f001:**
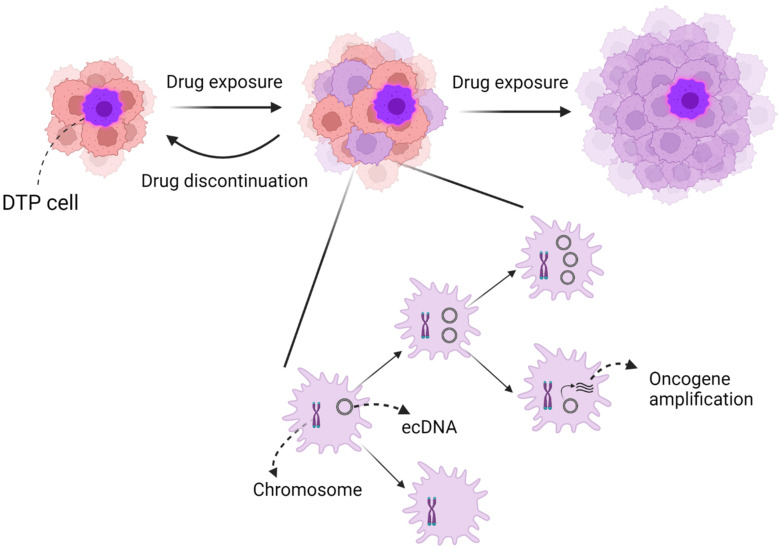
Image of intratumoral heterogeneity. Intratumoral heterogeneity, in particular, might consist of two mechanisms: (1) DTP cells and (2) chromosomal instability and ecDNA. DTP cells are similar to cancer stem cells and may also contribute to the evolution of tumor resistance. However, DTP cells may reverse to naïve cells after drug discontinuation, unlike cancer stem cells. In contrast, ecDNA may accelerate intratumoral heterogeneity more than chromosomal instability due to oncogene amplification and random dispersion to daughter cells.

**Table 2 ijms-24-04126-t002:** The rate of primary resistance in key clinical trials.

Study	EGFR-TKI	TKI-Generation	Overall Response Rate, %	Primary Resistance, %	Ref.
IPASS	Gefitinib	1	71.2	28.8	[4]
NEJ003	Gefitinib	1	73.7	26.3	[113]
WJTOG-3405	Gefitinib	1	62.1	37.9	[114]
EURTAC	Erlotinib	1	58	42	[6]
OPTIMAL	Erlotinib	1	83	17	[7]
LUX-Lung-7	Afatinib	2	72.5	27.5	[8]
ARCHER-1050	Dacomitinib	2	75	25	[9]
FLAURA	Osimertinib	3	80	20	[10]

## Data Availability

No new data were created or analyzed in this study. Data sharing is not applicable to this article.

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
