# Peer review of "Unraveling the Impact of Intratumoral Heterogeneity on EGFR Tyrosine Kinase Inhibitor Resistance in EGFR-Mutated NSCLC"

_ijms, 2023, doi:10.3390/ijms24044126_

Round 1

Reviewer 1 Report

The authors have performed a decent job compiling and presenting the relevant literature for the review. Overall, the manuscript is easy to understand and follow and presents a discussion on a topic of clinical and scientific importance. There are a few gaps and concerns which need to be addressed. Here are a few suggestions to further improve the quality and presentation of the review. Issues concerning the manuscript have been highlighted below:

1) The manuscript abstract claims that the discussion within the manuscript is centered around two perspectives: a) collateral signal pathway activation and b) intratumoral heterogeneity. However, the authors have not detailed the section on the collateral signal pathway activation. The entire section is short and hastened. The attention to detail is extremely limited. The authors need to provide more detail and review relevant literature outlining the details of multiple collateral signaling pathways.

2) The tumor microenvironment plays a pivotal role in the development of resistance to EGFR TKI treatment therapy in EGFR-mutated NSCLC. The authors need to include a section outlining the importance of changes in the tumor microenvironment and the local environmental factors that regulate the progression of NSCLC and the acquisition of resistance to TKIs.

3)  Integrins contribute to cancer progression and aggressiveness by activating intracellular signal transduction pathways and transducing mechanical tension forces. Remarkably, these adhesion receptors share common signaling networks with receptor tyrosine kinases (RTKs) and support their oncogenic activity, thereby promoting cancer cell proliferation, survival, and invasion. Integrins also play an important role in the development of resistance to therapies targeting RTKs and their downstream pathways. A remarkable feature of integrins is their wide-ranging interconnection with RTKs, which helps cancer cells to adapt and better survive therapeutic treatments. Additionally, EGFR has been shown to enhance integrin mechanics, cell spreading, FA organization, and maturation. Enhanced EGFR activity amplifies the mechanical phenotypes in cancer cells to promote cell behaviors that are associated with tumorigenic potential. Besides, more recently, integrin β3 has been shown to promote resistance to EGFR-TKI in NSCLC by upregulating the expression of AXL through the YAP pathway. The authors need to review these published sources and include the same in the current review.

a) Cruz da Silva E, Dontenwill M, Choulier L, Lehmann M. Role of Integrins in Resistance to Therapies Targeting Growth Factor Receptors in Cancer. Cancers (Basel). 2019 May 17;11(5):692. doi: 10.3390/cancers11050692. PMID: 31109009; PMCID: PMC6562376.

b) Rao TC, Beggs RR, Ankenbauer KE, Hwang J, Ma VP, Salaita K, Bellis SL, Mattheyses AL. ST6Gal-I-mediated sialylation of the epidermal growth factor receptor modulates cell mechanics and enhances invasion. J Biol Chem. 2022 Apr;298(4):101726. doi: 10.1016/j.jbc.2022.101726. Epub 2022 Feb 12. PMID: 35157848; PMCID: PMC8956946.

c) Sun Q, Lu Z, Zhang Y, Xue D, Xia H, She J, Li F. Integrin β3 Promotes Resistance to EGFR-TKI in Non-Small-Cell Lung Cancer by Upregulating AXL through the YAP Pathway. Cells. 2022 Jun 30;11(13):2078. doi: 10.3390/cells11132078. PMID: 35805163; PMCID: PMC9265629.

Author Response

The authors have performed a decent job compiling and presenting the relevant literature for the review. Overall, the manuscript is easy to understand and follow and presents a discussion on a topic of clinical and scientific importance. There are a few gaps and concerns which need to be addressed. Here are a few suggestions to further improve the quality and presentation of the review. Issues concerning the manuscript have been highlighted below:

We would like to thank reviewer 1 for our manuscript’s positive evaluation and constructive comments and suggestions that have helped us significantly improve it. Please, find next a detailed, point-by-point response to your comments.

1) The manuscript abstract claims that the discussion within the manuscript is centered around two perspectives: a) collateral signal pathway activation and b) intratumoral heterogeneity. However, the authors have not detailed the section on the collateral signal pathway activation. The entire section is short and hastened. The attention to detail is extremely limited. The authors need to provide more detail and review relevant literature outlining the details of multiple collateral signaling pathways.

We are most appreciative of your suggestion. We have taken care to incorporate additional details regarding the “collateral signaling pathways” in the section beginning at line 94 and ending at line 116 on page 3. Furthermore, we have made the necessary revisions to the abstract and title to accurately reflect our focus on the topic of intratumoral heterogeneity.

Notably, EGFR inhibition in NSCLC may result in the emergence of MET, HGF, and ERBB2 amplification, AXL activation, FGF2–FGFR1 loop mutations, IGF1R activation, and fusion events. MET amplification, which occurs in 4-10% of cases of resistance, stimulates downstream MEK/ERK and PI3K/Akt signaling pathways by phosphorylating ERBB3.[24-27] EGFR-mutant NSCLC patients may have MET amplification prior to EGFR-TKI treatment, and EGFR-TKI exposure may increase the rate of MET amplification and activate the bypass pathway through transactivated ERBB3.[28] HGF, which triggers the MET-HGF complex, also stimulates downstream MEK/ERK and PI3K/Akt signaling pathways through Gab1 as an adapter protein.[29] HGF overexpression has been shown to be commonly detected in NSCLC patients both before and after EGFR-TKI treatment (23/97, 24% and 45/97, 46%, respectively).[30] ERBB2 overexpression is a mechanism of acquired resistance for NSCLC patients without the EGFR T790M mutation after EGFR-TKI treatment, potentially initiated by ERBB3 which strongly activates downstream PI3K/Akt signaling.[31-35] This is be-cause ERBB3 can bind to the p85 subunit of PI3K and activate PI3K directly.[36]  AXL, activated by its ligand Gas6, can stimulate PI3K/Akt, MAPK, or NF-κB signaling and may also be associated with EMT.[37, 38] FGFR1 can form an autocrine loop with its ligand FGF2, activating downstream PI3/Akt signaling.[39]

IGF1R activation by IGF-IGF1R binding can also contribute to EGFR-TKI resistance, but its activation can be regulated by IGFBP-3 and IGFBP-4, which are active regulators of the six high-affinity IGFBPs (IGFBP-1 to IGFBP-6).[40-42]”

2) The tumor microenvironment plays a pivotal role in the development of resistance to EGFR TKI treatment therapy in EGFR-mutated NSCLC. The authors need to include a section outlining the importance of changes in the tumor microenvironment and the local environmental factors that regulate the progression of NSCLC and the acquisition of resistance to TKIs.

We are thankful for your suggestion. In response, we have included a section on the topic of tumor microenvironment between line 250 and line 274, which we hope will provide a complete understanding of the subject.

“3.2. Tumor microenvironment (TME)

Lung cancer cells interact with various cells in the tumor microenvironment (TME), including cancer-associated fibroblasts (CAFs) and tumor-associated macrophages (TAM), as well as normal cells, leading to intratumoral heterogeneity. CAFs, a major component of the TME, contribute to drug resistance through secretion of growth factors and chemokines, such as HGF.[81] HGF can activate MAPK and PI3K/AKT pathways through interaction with the MET receptor.[25, 26] A cross-talk between cancer cells and CAFs can also activate IGF-IGF1R pathway along with EMT- and stemness-related signaling.[82, 83] TAMs and myeloid-derived suppressor cells (MDSCs) also fill a pivotal role in resistance to EGFR-TKIs, with S100A9-positive MDSCs linked to poor response to EGFR-TKIs in NSCLC patients.[84]  Mechanistically, S100A9 upregulates ALDH1A1 expression and activates the retinoic acid (RA) signaling pathway, making the S100A9-ALDH1A1-RA axis a potential therapeutic target. Treatment with a pan-RAR antagonist can effectively inhibit cancer proliferation. [85]

Immunity

Drug exposure affects tumor and tumor-infiltrating immune cells. EGFR TKIs can lower PD-L1 expression in EGFR-mutant NSCLC cells, making PD-1/PD-L1 inhibitors ineffective for most patients with this type of cancer.[86] Furthermore, EGFR-mutant NSCLC cells with high PD-L1 expression are resistant to gefitinib due to PD-L1 over-expression that can trigger EMT through activation of the TGF-β/Smad pathway.[87] AXL has been linked to resistance to PD-1 inhibition and suppression of proper antigen presentation by the MHC-I.[88] Thus, inhibiting AXL could reorganize the immune-suppressive TME in a more favorable way.[89, 90]]”

3)  Integrins contribute to cancer progression and aggressiveness by activating intracellular signal transduction pathways and transducing mechanical tension forces. Remarkably, these adhesion receptors share common signaling networks with receptor tyrosine kinases (RTKs) and support their oncogenic activity, thereby promoting cancer cell proliferation, survival, and invasion. Integrins also play an important role in the development of resistance to therapies targeting RTKs and their downstream pathways. A remarkable feature of integrins is their wide-ranging interconnection with RTKs, which helps cancer cells to adapt and better survive therapeutic treatments. Additionally, EGFR has been shown to enhance integrin mechanics, cell spreading, FA organization, and maturation. Enhanced EGFR activity amplifies the mechanical phenotypes in cancer cells to promote cell behaviors that are associated with tumorigenic potential. Besides, more recently, integrin β3 has been shown to promote resistance to EGFR-TKI in NSCLC by upregulating the expression of AXL through the YAP pathway. The authors need to review these published sources and include the same in the current review.

  1. a) Cruz da Silva E, Dontenwill M, Choulier L, Lehmann M. Role of Integrins in Resistance to Therapies Targeting Growth Factor Receptors in Cancer. Cancers (Basel). 2019 May 17;11(5):692. doi: 10.3390/cancers11050692. PMID: 31109009; PMCID: PMC6562376.
  2. b) Rao TC, Beggs RR, Ankenbauer KE, Hwang J, Ma VP, Salaita K, Bellis SL, Mattheyses AL. ST6Gal-I-mediated sialylation of the epidermal growth factor receptor modulates cell mechanics and enhances invasion. J Biol Chem. 2022 Apr;298(4):101726. doi: 10.1016/j.jbc.2022.101726. Epub 2022 Feb 12. PMID: 35157848; PMCID: PMC8956946.

We are grateful for your suggestion. To acknowledge this evidence, we have added a few sentences regarding integrins in the section between line 123 and line 129 on page 3. We appreciate your input and hope that these additions are helpful.

“Another potential mechanism of collateral signal pathway activation is via integrins. Integrins are transmembrane receptors that facilitate cell-cell and cell-extracellular matrix adhesion and may play a vital role in countering EGFR-TKI treatments. Integrins and EGFR share similar signaling pathways, allowing cancer cells to adapt and withstand therapy more effectively.[46] Additionally, exposure to EGFR-TKIs can trigger the activation of integrin β3, boosting AXL expression through the YAP pathway and resulting in greater resistance.[47]”

Reviewer 2 Report

The authors discuss resistance mechanisms of EGFR-mutated NSCLC, by focusing on intratumoral heterogeneity in the manuscript. The topic itself has attracted much attention, owing to the importance of the development of novel anticancer agents.

<Major points>

1. In the "2. Collateral signal pathway activation" section, the authors just state several signal pathways concerning cell growth and proliferation. I suggest that the authors should make a figure summarizing the section content, by focusing on EGFR-related signaling, including its inhibitory mechanism.

2. Figure 1 seems too simple. The author should embody the more specific contents on extrachromosomal circular DNA that they mentioned in section 3.2.

3. Although this manuscript addresses scientific issues on resistance to EGFR kinase inhibitors, the resistance mechanism is not specifically and sufficiently described. Thus, the authors need to explain the resistance mechanism of the gatekeeper mutation like EGFR T790M including a structural figure in a separate section.

4. In Discussion, the author can introduce recent therapeutic strategies for the development of novel EGFR kinase inhibitors. Many medicinal chemists are trying inhibitor design, based on the inactivation conformation of A-loop in the EGFR kinase domain. Others suggest novel strategies for blocking the dimerization of the EGFR kinase domain.

<Minor points>

1. The title does not match with the manuscript content. Although the authors introduce two perspectives on resistance mechanisms of EGFR-mutated NSCLC, the title includes only one. In addition, the author should directly express what they think is "important" in the title.

2. In Abstract, the phrase "intertumoral heterogeneity" should be fixed to "intratumoral heterogeneity" (page 1, line 15).

3. In Introduction, the authors introduce several EGFR mutations. Please explain where their mutation sites are located in the EGFR structure, and how their mutations have an effect on EGFR function.

4. In Introduction, the authors introduce a total of three generation EGFR TKI. Please state the criterion for division of generation.

5. In Introduction, some references are missed. Please cite corresponding references (page 2, lines 50-52 and 72-74).

6. In Table 2, the author can add the respective chemical structures of inhibitors, along with approval years, approval countries, and respective references to the table.

Author Response

The authors discuss resistance mechanisms of EGFR-mutated NSCLC, by focusing on intratumoral heterogeneity in the manuscript. The topic itself has attracted much attention, owing to the importance of the development of novel anticancer agents.

We are grateful to Reviewer 2 for their favorable assessment of our manuscript and for providing insightful feedback and suggestions that have greatly enhanced its quality. In the following section, we have provided a comprehensive, itemized response to each of your remarks.

<Major points>

  1. In the "2. Collateral signal pathway activation" section, the authors just state several signal pathways concerning cell growth and proliferation. I suggest that the authors should make a figure summarizing the section content, by focusing on EGFR-related signaling, including its inhibitory mechanism.

We greatly appreciate your suggestion. To enhance the information, we have included additional details regarding collateral signaling pathways in the section ranging from line 94 to 116 on page 3. Furthermore, we have refined the abstract to emphasize our focus on intratumoral heterogeneity.

“Notably, EGFR inhibition in NSCLC may result in the emergence of MET, HGF, and ERBB2 amplification, AXL activation, FGF2–FGFR1 loop mutations, IGF1R activation, and fusion events. MET amplification, which occurs in 4-10% of cases of resistance, stimulates downstream MEK/ERK and PI3K/Akt signaling pathways by phosphorylating ERBB3.[24-27] EGFR-mutant NSCLC patients may have MET amplification prior to EGFR-TKI treatment, and EGFR-TKI exposure may increase the rate of MET amplification and activate the bypass pathway through transactivated ERBB3.[28] HGF, which triggers the MET-HGF complex, also stimulates downstream MEK/ERK and PI3K/Akt signaling pathways through Gab1 as an adapter protein.[29] HGF overexpression has been shown to be commonly detected in NSCLC patients both before and after EGFR-TKI treatment (23/97, 24% and 45/97, 46%, respectively).[30] ERBB2 overexpression is a mechanism of acquired resistance for NSCLC patients without the EGFR T790M mutation after EGFR-TKI treatment, potentially initiated by ERBB3 which strongly activates downstream PI3K/Akt signaling.[31-35] This is be-cause ERBB3 can bind to the p85 subunit of PI3K and activate PI3K directly.[36]  AXL, activated by its ligand Gas6, can stimulate PI3K/Akt, MAPK, or NF-κB signaling and may also be associated with EMT.[37, 38] FGFR1 can form an autocrine loop with its ligand FGF2, activating downstream PI3/Akt signaling.[39] IGF1R activation by IGF-IGF1R binding can also contribute to EGFR-TKI resistance, but its activation can be regulated by IGFBP-3 and IGFBP-4, which are active regulators of the six high-affinity IGFBPs (IGFBP-1 to IGFBP-6).[40-42]”

  1. Figure 1 seems too simple. The author should embody the more specific contents on extrachromosomal circular DNA that they mentioned in section 3.2.

We are grateful for your suggestion and apologize for any confusion regarding Figure 1. To clarify, the central importance of ecDNAs is that they do not follow Mendel's laws of inheritance, but rather are randomly distributed to daughter cells. This can result in intratumoral heterogeneity. Our aim is to demonstrate the disproportionate distribution of ecDNAs among cells by illustrating the number of ecDNAs in each cell in comparison to the conventional chromosomes. To further clarify, we have included an explanation in the Figure 1 legend as well.

Image of intratumoral heterogeneity. Intratumoral heterogeneity, in particular, might consist of two mechanisms; 1) DTP cells and 2) chromosomal instability and ecDNA. DTP cells are similar to cancer stem cells and may also contribute to the evolution of tumor resistance. However, DTP cells may reverse to naïve cells after drug discontinuation, unlike cancer stem cells. In contrast, ecDNA may accelerate intratumoral heterogeneity more than chromosomal instability due to oncogene amplification and dispersing randomly to daughter cells.”

  1. Although this manuscript addresses scientific issues on resistance to EGFR kinase inhibitors, the resistance mechanism is not specifically and sufficiently described. Thus, the authors need to explain the resistance mechanism of the gatekeeper mutation like EGFR T790M including a structural figure in a separate section.

We are grateful for your suggestion. To address the issue, we have included a sentence to the resistance mechanisms based on the structural changes of T790M in the section ranging from Line 170 to 171 on page 6 for your review.

A conformational shift in the kinase known as EGFR T790M lowers drug affinity for the target.”

  1. In Discussion, the author can introduce recent therapeutic strategies for the development of novel EGFR kinase inhibitors. Many medicinal chemists are trying inhibitor design, based on the inactivation conformation of A-loop in the EGFR kinase domain. Others suggest novel strategies for blocking the dimerization of the EGFR kinase domain.

We appreciate your suggestion and understand that your thoughts are valuable. We have taken the liberty of incorporating your suggestion into the discussion section, specifically between lines 453 and 462, regarding the therapeutic strategy of fourth-generation EGFR-TKIs.

“Recently, the development of fourth-generation EGFR-TKIs has gained traction.[18, 19] These drugs target resistance mechanisms, such as the EGFR C797X mutations commonly seen in patients who have developed resistance to third-generation EGFR TKIs. Despite their promising potential, these TKIs still face several challenges in effectively targeting on-target resistance mechanisms. One of the biggest obstacles is the potential for the emergence of drug resistance, which could result from off-target mutations or alternative signaling pathway activation. Furthermore, the optimal method of using fourth-generation EGFR TKIs in conjunction with other treatments, such as immunotherapy or chemotherapy, is yet to be determined and requires further research to fully understand their ability to manage intratumoral heterogeneity.”

<Minor points>

  1. The title does not match with the manuscript content. Although the authors introduce two perspectives on resistance mechanisms of EGFR-mutated NSCLC, the title includes only one. In addition, the author should directly express what they think is "important" in the title.

We are grateful for your suggestion and have made alterations to the title and abstract accordingly. Our focus is now centered on elaborating on the subject of intratumoral heterogeneity. The new title of our work is “Unraveling the Impact of Intratumoral Heterogeneity on EGFR Tyrosine Kinase Inhibitor Resistance in EGFR-Mutated NSCLC.

  1. In Abstract, the phrase "intertumoral heterogeneity" should be fixed to "intratumoral heterogeneity" (page 1, line 15).

We are thankful for your suggestion and have taken it into account. The necessary changes have been made as per your request.

  1. In Introduction, the authors introduce several EGFR mutations. Please explain where their mutation sites are located in the EGFR structure, and how their mutations have an effect on EGFR function.

Thank you for your suggestion. In response, we have incorporated the details of EGFR mutations between lines 41 and 48 on page 2.

Approximately 90% of the most commonly occurring mutations in the EGFR gene are the classical activating mutations of exon 19 deletions and exon 21 L858R point mutations, which lead to the continuous activation of EGFR.[11] The most frequently observed uncommon EGFR mutations include G719X, S768I, L861Q, exon 20 insertions, and compound mutations. Uncommon EGFR mutations are any other mutations in the EGFR gene that are not as commonly seen as the classical mutations. These mutations may have differing effects on the function of the EGFR protein and may also respond differently to EGFR-targeted drug treatments.[12]

  1. In Introduction, the authors introduce a total of three generation EGFR TKI. Please state the criterion for division of generation.

We are appreciative of your suggestion. To incorporate your thoughts, we have included the relevant sentences in the section ranging from line 51 to 54 on page 2 for your review.

“The "second-generation" EGFR-TKIs earned their name due to their irreversible pan-HER inhibitory activity. "Third-generation" inhibitors target the T790M mutation, which is the most frequent resistance mechanism to both first and second-generation EGFR-TKIs, specifically.[13]”

  1. In Introduction, some references are missed. Please cite corresponding references (page 2, lines 50-52 and 72-74).

We are grateful for your suggestion. In response, we have added references #4-10 to the section between lines 50 and 52 in the last version. Additionally, we have included reference #21 in the section between lines 72 and 74 in the same version.

“Many studies have revealed several acquired-resistance mechanisms and developed therapeutic strategies countervailing them.[4-10]”

“Previous studies have revealed that suppressing the main driver oncogene signaling pathway can lead to the complementary activation of other collateral bypass signaling pathways.[22]”

  1. In Table 2, the author can add the respective chemical structures of inhibitors, along with approval years, approval countries, and respective references to the table.

We are grateful for your thoughtful suggestion. We have made a note of the key references and meticulously reorganized the reference numbers. We have decided not to include the approval years or countries, given the heterogeneity in regulatory approvals in many countries worldwide, and to keep the review broad and applicable to a diverse readership. Once again, thank you for your invaluable contribution.

Round 2

Reviewer 2 Report

The manuscript has been quite improved; therefore I recommand the acception of this manuscript for publication.